# Nanoassemblies from Amphiphilic Sb Complexes Target Infection Sites in Models of Visceral and Cutaneous Leishmaniases

**DOI:** 10.3390/pharmaceutics14081743

**Published:** 2022-08-21

**Authors:** Juliane S. Lanza, Virginia M. R. Vallejos, Guilherme S. Ramos, Ana Carolina B. de Oliveira, Cynthia Demicheli, Luis Rivas, Sébastien Pomel, Philippe M. Loiseau, Frédéric Frézard

**Affiliations:** 1Faculty of Pharmacy, Antiparasite Chemotherapy, UMR 8076 CNRS BioCIS, University Paris-Saclay, F-92296 Chatenay-Malabry, France; 2Department of Physiology and Biophysics, Institute of Biological Sciences, Universidade Federal de Minas Gerais, Belo Horizonte 31270-901, MG, Brazil; 3Department of Chemistry, Institute of Exact Sciences, Universidade Federal de Minas Gerais, Belo Horizonte 31270-901, MG, Brazil; 4Centro de Investigaciones Biológicas Margarita Salas-CSIC, 28040 Madrid, Spain

**Keywords:** leishmaniasis, nanoassemblies, antimony, drug targeting, amphiphilic complexes, miltefosine, leishmania

## Abstract

This work aims to evaluate whether nanoassemblies (NanoSb) made from antimony(V) complexes with octanoyl-*N*-methylglucamide (SbL8) or decanoyl-*N*-methylglucamide (SbL10) would effectively target the infection sites in visceral and cutaneous leishmaniases (VL and CL). NanoSb were investigated regarding stability at different pHs, accumulation of Sb in the macrophage host cell and liver, and in vitro and in vivo activities in models of leishmaniasis. The kinetic stability assay showed that NanoSb are stable at neutral pH, but release incorporated lipophilic substance after conformational change in media that mimic the gastric fluid and the parasitophorous vacuole. NanoSb promoted greater accumulation of Sb in macrophages and in the liver of mice after parenteral administration, when compared to conventional antimonial Glucantime^®^. SbL10 was much more active than Glucantime^®^ against intramacrophage *Leishmania* amastigotes and less cytotoxic than SbL8 against macrophages. The in vitro SbL10 activity was further enhanced with co-incorporated miltefosine. NanoSb showed high antileishmanial activity in the *L. donovani* murine VL after parenteral administration and moderate activity in the *L. amazonensis* murine CL after topical treatment. This study supports the ability of NanoSb to effectively deliver a combination of Sb and co-incorporated drug to host cell and infected tissues, in a better way than Glucantime^®^ does.

## 1. Introduction

Leishmaniases are a complex of infectious diseases, including visceral and cutaneous forms, caused by different species of *Leishmania* protozoa and transmitted to mammals by the bite of a phlebotomine sandfly vector. Those are associated with poor housing, lack of financial resources, population displacement, malnutrition, and a weak immune system, leading to their classification as neglected tropical diseases [1]. Chemotherapy is essential to treat and control visceral (VL) and cutaneous (CL) leishmaniases, as there is no vaccine approved for human use [1]. Pentavalent antimony (Sb(V)) complexes, despite their use for more than 70 years, are still used as first-choice drugs in several developing countries for treatment of all forms of leishmaniases. These include sodium stibogluconate (Pentostam^®^) and meglumine antimoniate (Glucantime^®^, Sanofi-Aventis Farmacêutica Ltda, São Paulo, Brazil). Pentavalent antimonials are prodrugs in which the metal is reduced to the active and toxic trivalent form [2]. Unfortunately, the use of conventional antimonial drugs is limited by the need for parenteral administration through repeated doses for a long period of time, by severe toxicities, and the development of drug resistance [2]. The few other drugs available, including amphotericin B (AmB) and miltefosine (Milt), also present drawbacks, mainly toxicities and emergence of resistance. Thus, there is a great need for new drugs and novel delivery strategies to improve existing drugs [3].

Amphiphilic Sb(V) complexes were first introduced by our group to achieve orally effective pentavalent antimonials [4]. Two complexes, named SbL8 and SbL10, were obtained through reaction of KSb(OH)_6_ with the non-ionic surfactants, *N*-octanoyl-*N*-methylglucamide (L8) and *N*-decanoyl-*N*-methylglucamine (L10), at a molar ratio of 1:3, respectively. A pharmacokinetic study of SbL8 by oral route in mice showed greater and more sustained levels of Sb in serum and liver when compared with Glucantime^®^, resulting in area under the curve (AUC) and mean residence time (MRT) about 4-fold greater [4,5]. SbL8 further demonstrated antileishmanial activities by the oral route in murine models of VL and CL caused by New World *Leishmania* species [4,5,6]. SbL8 self-assembles in aqueous solution, forming micelle-like nanoparticles capable of incorporating other lipophilic substances [5,6]. Nevertheless, the therapeutic potential of these nanosystems for carrying and delivering Sb(V) and other co-incorporated substances has not been fully investigated, mainly regarding their use by parenteral and topical routes. Indeed, nanoparticles have been extensively studied as drug carrier for treatment of VL by parenteral route, exploiting their natural tendencies to accumulate in the macrophages of the liver, spleen and bone marrow, which are the main infection sites [3,7]. Nanoassemblies are also promising in topical formulations for CL, due to improved drug solubility and penetration into the skin, when compared to conventional formulation [8,9].

In the present work, we tested the hypothesis as to whether SbL8 and SbL10 nanoassemblies may favor drug targeting to the infection sites after parenteral and topical administrations. To achieve this goal, we evaluated the thermodynamic and kinetic stabilities of these nanoassemblies at different pHs and investigated their ability to promote Sb uptake by macrophages and the liver, as well as their in vitro antileishmanial activities and therapeutic efficacies in murine models of VL and CL.

## 2. Materials and Methods

### 2.1. Reagents, Culture Media and Drugs

M199, αMEM, RPMI-1640 culture media, adenosine, NaHCO_3_, NaOH, HEPES, penicillin-streptomycin solution, dimethyl sulfoxide (DMSO) and phorbol 12-myristate 13-acetate (PMA) reagent grade, Giemsa stain, *N*-octanoyl-*N*-methylglucamide (L8, 98%), *N*-decanoyl-*N*-methylglucamide (L10, 98%), potassium hexahydroxoantimonate (KSb(OH)_6_), potassium antimony(III) tartrate, 1,6-diphenylhexatriene (DPH), miltefosine (Milt) powder (≥98%) and amphotericin B (AmB) (European Pharmacopoeia (EP) Reference Standard) were purchased from Sigma-Aldrich Co. (St. Louis, MO, USA). Trypan blue was from Merck. Hemin was obtained from Honeywell International Inc. (Charlotte, North Carolina, USA) and fetal calf serum (FCS) was supplied by Invitrogen or Cultilab (Brazil) and inactivated by heating at 56 °C for 1 h (HIFCS). Propylene glycol (PG) was obtained from Labsynth (Diadema, Brazil). Double-distilled deionized water was used throughout all the experiments. Glucantime^®^ was from Sanofi-Aventis Farmacêutica Ltda (São Paulo, Brazil). The fluorescent analog of miltefosine (MT-11-BDP) (11-(4′,4′-difluoro-1′,3′,5′,7′-tetramethyl-4′-bora-3′a,4′a-diaza-s-indacen-2′-yl)undecylphosphocholine) was synthesized as described previously [10].

### 2.2. Synthesis of Amphiphilic Antimony Complexes

The amphiphilic complexes SbL8 and SbL10 were synthesized, as previously described [4]. Briefly, KSb(OH)_6_ and L8 or L10 surfactant were co-dissolved in water at a 1:3 Sb/surfactant molar ratio and a final surfactant concentration of 0.08 M. The mixture was heated at 60 °C under agitation until complete solvent evaporation. The resulting film was dissolved in water at 25 °C, and the dispersion was finally freeze-dried.

### 2.3. Preparation of the Formulations for Biological Assays

To prepare the formulations for in vitro assays and parenteral administration, the freeze-dried compounds were first dispersed in 0.15 M NaCl solution at 175 mM Sb concentration. Incorporation of Milt into SbL10 suspension was performed as described previously, through simple addition of Milt to the SbL10 suspension and incubation for 1 h at room temperature [6].

To prepare formulations for topical use, the freeze-dried compounds were dispersed in a 1:1 (*v*/*v*) water:PG mixture at 12% Sb (*w*/*v*) and hydroxyethyl cellulose (Natrosol™, Ashland, Catlettsburg, KY, USA) was dispersed in the resulting solution at 1% (*w*/*v*) final concentration.

### 2.4. Evaluation of Incorporation of Miltefosine Fluorescent Analog

MT-11-BDP was used as a fluorescent indicator of the incorporation of Milt in SbL10 nanoassemblies, as previously described [5]. An aliquot of 10 mM ethanolic solution of the MT-11-BDP was added to the bottom of a tube and the solvent was evaporated. The dispersion of SbL10 or L10 in saline was then added to reach an Sb/MT-11-BDP molar ratio of 130:1, followed by incubation for 1 h at 25 °C. A saline solution of MT-11-BDP without surfactant was used as control. For fluorescence measurement, the solutions were diluted in PBS at an L8 concentration of 1 mM. Fluorescence measurements were carried out using the Cary Eclipse™ spectrofluorometer (Varian Inc., Australia) and a 1-cm cuvette compartment with temperature control and magnetic stirring. Fluorescence emission spectra were recorded at 25 °C, with excitation wavelength (λ) set at 500 nm.

### 2.5. Physicochemical Characterization of SbL8 and SbL10 Nanoassemblies

The formulations were characterized for particle size distribution (mean hydrodynamic diameter and polydispersity index) by dynamic light scattering (DLS) and zeta-potential, using the Zetasizer equipment (Nano ZS90; Malvern Instruments, Malvern, UK) and measurements at a fixed angle of 90°. Dispersion Technology Software, version 6.12, was used to collect and analyze the data obtained. SbL8 and SbL10 dispersions were diluted in 0.15 M NaCl solution at final concentration of 30 mM of Sb, and kept at 25 °C during the entire experiment. The particle size was also investigated using nanoparticle tracking analysis (NTA; Malvern Instruments, UK) and NTA 3.44 software (Malvern Panalytical, Malvern, UK) to collect and analyze data. Measurements were performed at a final Sb concentration of 1 mM in phosphate buffer saline (PBS, 0.15 M NaCl, 10 mM phosphate) at pH 7.2.

The formation of hydrophobic environment in aqueous dispersions of L8 and SbL8 was investigated using the lipophilic fluorescent probe DPH, as described previously [4], exploiting the increase in DPH fluorescence upon incorporation into hydrophobic environment. To evaluate the dependence of DPH fluorescence on SbL8 concentration, DPH was added from a tetrahydrofuran stock solution at a final concentration of 5 × 10^−7^ M to dispersions of SbL8 at various concentrations in either water (pH 5.5), HCl 0.05 M or PBS at pH 5.8 or 7.2. After 24-h incubation at 25 °C under light protection, the fluorescence intensity was measured at excitation and emission wavelengths of 360 and 428 nm, respectively. Fluorescence measurements were carried out using a Cary Eclipse™ fluorescence spectrometer. Temperature was controlled at 25 °C through a jacketed cuvette holder from a refrigerated circulating water bath.

To evaluate the kinetic stability of SbL8 nanoassemblies, a dispersion of SbL8 was prepared at 50 mM L8 in deionized water and incubated overnight with DPH at 10^−5^ M final concentration. The suspension was diluted 50 times in either HCl 0.05 M, PBS pH 7.2, 5.8 or 4.5 in quartz cuvette maintained at 37 °C under magnetic stirring. To evaluate the kinetic stability of SbL10 nanoassemblies, a dispersion of SbL10 was prepared at 50 mM of L10 in deionized water and further incubated overnight at 25 °C with 10^−4^ M DPH final concentration. The SbL10 suspension was then diluted at 0.1 mM L10 in either HCl 0.05 M, PBS 7.2, 5.8 or 4.5 in a quartz cuvette maintained at 37 °C under magnetic stirring. DPH fluorescence was then registered continuously as a function of time at excitation and emission wavelengths of 360 and 428 nm, respectively. The half-times of dissociation of the nanoassemblies were determined through non-linear regression fit according to mono-exponential decay model, using the GraphPad Prism software^©^, Version 9, GraphPad Software, LLC (San Diego, CA, USA).

Small-angle X-ray scattering (SAXS) studies were conducted in line 2 of LNLS—Brazilian Synchrotron Light Laboratory/MCT (Campinas, Sao Paulo, Brazil). This line is equipped with a monochromator (λ = 1.54 Å), an ionization chamber, and a 300 k Pilatus detector placed at 1 m from the sample to record the intensity of the scattering. The scattering from the samples was subtracted from that of the system without the sample. The intensities of all samples were measured in relative units, but for the purpose of comparison, the measurements were standardized under the same experimental conditions. A volume of 50 µL of SbL8 dispersions in either water, HCl 0.05 M or PBS at pH 7.2 was injected into the sample compartment maintained at 25 °C. The I(q) vs. q curves were obtained after subtraction of the corresponding blank from the signal of each sample.

Transmission electron microscopy (TEM) was also used to characterize the morphology of SbL8 nanoassemblies. A solution of SbL8 at 25 mM was prepared in PBS 7.2. The sample was deposited on formvar film (previously deionized copper grid) and treated with osmium tetroxide (OsO_4_) as contrast agent. The images were obtained using a Tecnai G2-Spirit-FEI-2006 (120 kV) microscope (FEI Company, Hillsboro, OR, USA), equipped with high resolution (11 megapixels) and high-speed digital cameras, located at the Electron Microscopy Center of UFMG.

### 2.6. Antimony Intracellular Accumulation

The intracellular accumulation of Sb was evaluated in THP-1 acute monocytic human leukemia cell line (ATCC TIB-202) using 6-wells plates. In each well, 2 × 10^6^ THP-1 cells were differentiated into macrophage-like, adherent and non-dividing phenotype, via treatment with 100 ng/mL PMA for 72 h in complete RPMI-1640 medium. Cells were then exposed for 4 h at 37 °C to the antimonial drugs (Glucantime^®^, SbL8 and SbL10) at 50 µg Sb/mL in RPMI-1640 medium complemented with 10% HIFCS (*n* = 6–9, for each antimonial drug). Adherent cells were then washed 4 times with 2 mL of ice-cold PBS and digested overnight with 65% nitric acid at room temperature. The digested samples were then diluted 50-fold with 0.2% nitric acid and Sb concentration was determined by graphite furnace atomic absorption spectrometry (GFAAS), using a Perkin–Elmer AA600 graphite furnace atomic absorption spectrometer, as described previously [11]. Cells without drug exposure were used for blank subtraction.

### 2.7. Hepatic Accumulation of Antimony

BALB/c mice (female, 6 to 8 weeks old, 18 to 20 g) were obtained from the Animal Facility Center of the Federal University of Minas Gerais (UFMG). Free access to a standard diet was allowed, and tap water was supplied ad libitum. The animals were handled according to the protocol approved by the Ethical Committee for Animal Experimentation of the UFMG (protocol no. 163/2019). Five groups of BALB/c mice (*n* = 5) were used for the biodistribution study of Sb. They received a single dose of each compound either by i.p. or oral route and antimony level was determined in the liver 24 h after administration. The compounds were Glucantime^®^ (200 mg Sb/kg, i.p. route), SbL8 or SbL10 (20 mg Sb/kg, i.p. route), or SbL8 and SbL10 (200 mg Sb/kg, oral route). Animal euthanasia was undertaken by cervical dislocation after ketamine-xylazine anesthesia. The livers were recovered, homogenized, and subjected to digestion with nitric acid in a dry block (MA 4004; Marconi, São Paulo, Brazil). Antimony concentration was determined in digested liver by GFAAS using a Perkin–Elmer AA600, as described previously [11]. The analytical method for determination of Sb in the liver was validated and showed suitable levels of precision (coefficient of variation [CV] < 5%), accuracy (80 to 120% analyte recovery), and linearity (range of 10 to 180 µg of Sb/L). The quantification limit of the analytical method was 0.93 µg of Sb/g for the liver.

### 2.8. In Vitro Cytotoxicity and Antileishmanial Activity

RAW264.7 (mouse leukemic monocyte macrophage) cell line from the European Collection of Cell Cultures (UK) was used for the cytotoxicity assay and for *Leishmania* infection to determine activity against intracellular amastigotes. RAW264.7 cell were kept in 75-cm^2^ sterile flasks in a total volume of 20 mL of RPMI-1640 medium complemented with 10% HIFCS, at 37 °C in 5% CO_2_ atmosphere incubator. Cells were passed every 4 days (log phase) at 100,000 cells per mL of inoculum.

Promastigotes forms of *Leishmania* (L.) *donovani* (MHOM/ET/67/HU3, also called LV9) were cultivated in M199 medium plus 10 mM of HEPES pH 7 supplemented with adenosine 100 µM and hemin at 0.5 mg/L, 10% FCS, 100 I.U./mL penicillin and 100 µg/mL streptomycin, pH 7, as adapted from a previous study [12]. Cells were passed in fresh medium every 4 days (log phase) at 200,000 parasites per mL in a final volume of 5 mL kept into 25 cm^2^ bottles at 27 °C (+/− 2 °C) incubator. A maximum number of 10 passages from primary culture of promastigotes isolated from spleen of experimentally infected rodents were used for all experiments.

For cytotoxicity test, RAW264.7 cells were resuspended in 20 mL of fresh medium and counted with Trypan blue at 0.4% (*w*/*v*) in Neubauer chamber. Cell suspension was adjusted to 100,000 cells/mL and aliquoted into a sterile 96-well plate (200 µL/well). Plates were incubated for 24 h at 37 °C, 5% CO_2_. Afterwards, supernatants were discarded and 100 µL of new medium containing serial dilutions of SbL8, SbL10, free ligands L8 and L10, Glucantime^®^, AmB, Milt, potassium antimony(III) tartrate were distributed in triplicate for each concentration. Plates were incubated for 24 h in the same conditions as described above. After this period, plates were centrifuged, supernatants were removed and 100 µL of new medium containing 100 µM resazurin, were added to each well. Plates were further incubated for 4 h and fluorescence intensities (FI) were read with Spark^®^ Tecan Spectrometer (Männedorf, Switzerland, EU) at 530–570 nm excitation and 585–590 nm emission wavelength. CC_50_ (cytotoxic concentration for 50% of cells) were calculated for each drug, taken DMSO at 10% (*v*/*v*) and RPMI-1640 medium supplemented with 10% HIFCS as negative (0%) and positive (100%) controls for viability, respectively.

Axenic amastigote-like forms were obtained and used as previously described [13]. Briefly, stationary phase cultures of LV9 were centrifuged at 500× *g* for 10 min, and parasite pellet was resuspended into fresh complete M199 containing 200 µM of Ca^2+^ and Mg^2+^ at pH 6. Cells were incubated at 37 °C in 5% CO_2_ atmosphere for 72 h before drug screening. After incubation, plates were centrifuged, and supernatant was discarded and substituted by serial dilutions of aforementioned drugs (100 µL/well). After 24 h of incubation in the same above conditions, resazurin solution was added at final concentration of 100 µM and plates were incubated for additional 24 h. Fluorescence was read as above and IC_50_ (inhibitory concentration of 50%) were determined for each drug, taken as 100% and 0% viability the values measured for untreated and 10% DMSO-treated amastigote-like, respectively.

The assay on intracellular amastigotes was performed as described previously [13,14]. Briefly, a total of 20.000 RAW264.7 cells in 200 µL of fresh complete RPMI plus 10% HIFCS, cultivated as described above, were added to each well in a Nunc™ Lab-Tek™ Chamber Slide System (Thermo Fisher Scientific™, Waltham, MA, USA) and incubated at 37 °C in 5% CO_2_ for 24 h before *Leishmania* infection. Axenic LV9 amastigote-like forms (obtained as described above) were added at 10:1 parasites/RAW264.7 cell ratio in fresh complete RPMI 10% HIFCS and incubated for 24 h in the same conditions to allow parasite infection. After this period, wells were washed twice and fresh medium containing serial dilutions of test drugs was added (100 µL/well). Plates were incubated for additional 24 h. Wells were then washed three times with PBS, fixed with ice-cold methanol (>99%) for 2 min, stained for 5–10 min with 10% aqueous solution of Giemsa stain, washed in water and air-dried for further microscope analysis. At least 300 macrophages were counted per well, regardless of whether those were infected or not. Only assays in which the percentage of infected macrophages was higher than 80% were considered. The infection index was defined as II = (% infection rate × amastigotes/infected macrophage)/100. The IC_50_ in this case is the drug concentration in which the II corresponds to 50%, using the non-treated control as 100% [14].

Both IC_50_ and CC_50_ values in all experiments were calculated in GraphPad Prism^®^ 9.2.0 software through non-linear regression of log(inhibitor) vs. response (variable slope). Only results in each Hillslope was around −3 and R Square was >0.95 were considered. Results are from a representative experiment out of at least three independent experiments.

### 2.9. Chequerboard Assay

In order to explore possible pharmacological interactions (synergism, additive effect or antagonism) between the clinically used leishmanicidal compounds Milt and the novel amphiphilic Sb complex SbL10 described in this work, a chequerboard assay was performed. After determining the IC_50_ of each compound itself in LV9-infected macrophages, as described above the top concentration for each compound, set as 4 × IC_50_ (Milt or SbL10)/5 were calculated and different combinations of Milt in SbL10 aqueous solution were prepared based on subsequent variations of each top concentration at: (A) 0.2:0.8; (B) 0.4:0.6; (C) 0.6:0.4 and (D) 0.8:0.2 ratios before serial dilutions in a two-dimensional concentration array, as described and adapted from previous studies [12,15,16,17]. Serial dilutions of each compound alone were also tested. RAW264.7 were plated in a 96-well plate and infected with amastigote-like forms of LV9, as described in the previous topic. A total of 24 h after infection, washing and cell-resting, 100 µL of drug solutions were distributed per well and incubated 48 h at 37 °C under 5% CO_2_ atmosphere. At the end of the incubation, cells were lysed and DNA was extracted by adding 50 µL of DirectPCR^®^ Lysis Reagent (Viagen Biotech Inc, Eurogentec, France) with Proteinase K (1 mg/mL), and plates were further incubated for 2 h at 4 °C, according to manufacturers’ information. Parasite load in cells was determined using SYBR Green I (Invitrogen, France) assay, as described previously [12] in which the fluorescence dye is shown to be enhanced upon parasite DNA coupling. A total of 40 µL of lysis buffer containing 0.05% of SYBR^®^ Green (5 μL of SYBR Green I/10 mL probe) and 10 µL of cell lysate were used per reaction. Untreated, non-infected and infected macrophages were used as controls. The Mastercycler^®^ ep realplex real-time PCR machine (Eppendorf, France) was used to directly evaluate the fluorescence, after the application of the following program: 90 °C for 1 min and ramp time of 5 min until 10 °C, fluorescence obtained in continue and in a hold step at 10 °C. The fluorescence values at 10 °C were analyzed to determine the IC_50_ for each drug and its combination. The IC_50_ for each association was then used to calculate the fractional inhibitory concentration (FICs) which corresponds to IC_50_ of molecule A when mixed with molecule B/IC_50_ of molecule A alone [15].

### 2.10. Antileishmanial Chemotherapy in Murine VL Model

The in vivo experiment and animal handle described here were carried out according to European Directive 2010/63/EU, the article R214.89 of the French Animal Welfare Law n˚2013–1118. The specific protocols for in vivo testing of antileishmanial potential drugs were approved by the Paris-Saclay University’s institutional ethic committee for animals handling, license CEEA 26-063/2013 and followed previously described studies [18,19]. Golden hamsters (*Mesocricetus auratus*) previously infected with *L. donovani* (LV9) were euthanized and spleens were processed to obtain viable amastigotes for mice infection. BALB/c mice (Janvier Labs, France), females, aged from 6 to 8 weeks were then infected with an inoculum of 5 × 10^6^ parasites, by retro-orbital intravenous injection. One week after infection, mice received different treatments per group (*n* = 6–9) once a day for 30 days, in the following schemes: a. Untreated control; b. Glucantime^®^ (200 mg Sb/kg/day; i.p. route; c. SbL8 (20 mg Sb/kg/day; i.p.); d. SbL10 (20 mg Sb/kg/day; i.p.); e. SbL8 at 200 mg Sb/kg/day, by oral route, and f. SbL10 (200 mg Sb/kg/day; oral route). After the 30-days treatment, mice were euthanized, and their liver and spleen were analyzed for determination of parasite load by qPCR as described below (Section 2.12).

### 2.11. Antileishmanial Chemotherapy in Murine CL Model

The *Leishmania* strain used for cutaneous infection *Leishmania* (L.) *amazonensis* (MHOM/BR/1989/BA199) was obtained from the cryopreservation bank of the *Leishmania* Biology Laboratory at ICB, UFMG. Cells were maintained in vitro as promastigotes at 24 °C, pH 7.0, in αMEM supplemented with 10% HIFCS, 100 μg/mL kanamycin, and 50 μg/mL, ampicillin in BOD incubator. The promastigotes were grown in cell culture flasks of 25-mL with initial inoculum of 1 × 10^6^ cells/mL and transferred to fresh medium, twice a week, once they reached the stationary growth phase.

BALB/c mice (female, 6 to 8 weeks old, 18 to 20 g) were obtained from the Animal Facility Center of the Federal University of Minas Gerais (UFMG). Free access to a standard diet was allowed, and tap water was supplied ad libitum. The animals were handled according to the protocol approved by the Ethical Committee for Animal Experimentation of the UFMG (protocol no. 54/2020). Mice were first inoculated intradermally at the tail base with 1 × 10^6^ stationary phase promastigotes of *L. amazonensis*. Chemotherapy was initiated 35 days post-infection, corresponding to the first ulceration sign of the infection papule, with daily doses for 30 days. Animals were divided in groups of 6 individuals receiving the following treatments: a. topical application of 50 µL of 1:1 water:PG hydrogel containing SbL8 at 12% Sb (*w*/*v*); b. topical application of 50 µL of 1:1 water:PG hydrogel containing SbL10 at 12% Sb (*w*/*v*); c. topical application on the lesion of 50 µL of 1:1 water:PG hydrogel containing Glucantime^®^ at 12% Sb (*w*/*v*); i.p. administration of Glucantime^®^ at 200 mg Sb/Kg/day; d. non-treated control group. After chemotherapy (65 days post-infection), animals were euthanized and the lesion was removed for evaluation of the parasitic load by qPCR, as described below (Section 2.12).

### 2.12. Evaluation of Parasite Load by qPCR

Evaluation of anti-leishmanial activity in the VL and CL murine models were performed at the end of each chemotherapeutic protocol, as described previously [5,6]. Animals were submitted to euthanasia to collect and analyze organs (liver and spleen in VL; excision of the whole lesion in CL). Organs were weighed and homogenized in cold PBS (1 mL per 100 mg of tissue) with the aid of a tissue dissociator. One hundred microliters of each organ homogenate were added in a microtube containing lysis buffer and Proteinase K, vortexed and incubated at 56 °C overnight to extract genomic DNA according to manufacturers’ instructions, with the “Illustra tissue & cells genomic Prep Mini Spin Kit” (GE, Healthcare Lifesciences) or NucleoSpin^®^ Tissue Kit (MN, Macherey-Nagel GmbH & Co. KG, Dürin, Germany). Likewise, genomic DNA was extracted from 1 million axenic amastigote-like of *L. donovani* (LV9) or *L. amazonensis* promastigotes, and used for serial dilutions defining the standard curve as absolute quantitative control. The DNA concentrations were measured by spectrophotometry (Abs at 280/260 nm) and adjusted to 50 ng/μL per reaction. The final volume was 20 μL per reaction that included ultrapure water, SYBR Green PCR Master Mix (Warrington, UK), 10 pmol of each oligonucleotide, as sense (forward, 5′-CCTATTTTACACCAACCCCCAGT-3′) and antisense primers (reverse, 5′-GGGTAGGGGCGTTC TGCGAAA-3′) constructed for amplification of the mini-circle region present in the kinetoplast DNA (kDNA) of approximately 120 bp. Real-time qPCR was performed as described previously [6,18,19]. The amplification protocol included an annealing temperature and extension of 60 °C, with melting curve construction, on the Applied Biosystems™7500 Fast Real-Time PCR System (Thermo Fisher Scientific, Waltham, MA, USA) and the analysis was made using the 7500 System Software. Parasite load was determined by absolute quantification based on the standard curve Ct vs. gDNA mass in pg. Results are shown as the parasite load that represents the number of *Leishmania* per ng of total DNA.

Additionally, in the cutaneous infection model, lesion development was followed during the treatment and plot of the change in lesion size with respect to time zero was obtained as a function of time. As the lesions developed into a circular shape, the size was calculated from the means of the coronal and sagittal lesion diameters, using a universal calliper, 150 mm, Digimess (São Paulo, Brazil), every 5 days from the onset of the treatment.

### 2.13. Statistical Analyses

One-way ANOVA with Tukey’s post-test (for normally distributed data) or Kruskal–Wallis non-parametric test with Dunn’s post-test were used for statistical analyses of parasite load, with significance level *p* < 0.05. The normal distribution was checked with the following tests: Anderson-Darling test, D’Agostino & Pearson test and Shapiro–Wilk test. Two-way ANOVA (repeated measures) was used to compare the variation in lesion size between the experimental groups, followed by Dunnett’s post-test. *p* < 0.05, *p* < 0.01, *p* < 0.001 and *p* < 0.0001 were marked with *, **, *** and ****, respectively. The graphics and statistical analyses were performed using GraphPad Prism^®^ (version 9) software (GraphPad Software LLC, San Diego, CA, USA).

## 3. Results

### 3.1. Effect of pH on the Stability of SbL8 and SbL10 Nanoassemblies

Previous studies led to the identification of 1:3 Sb-ligand complexes as the main species in SbL8 and SbL10 compounds [4], as illustrated in Figure 1.

The formation of micelle-type nanoassemblies with a spherical core–shell was also established for SbL8 using dynamic light scattering (DLS), nanoparticle tracking analysis (NTA), transmission electron microcopy (TEM), atomic force microscopy (AFM) and Small-angle X-ray scattering (SAXS) [4,5]. SbL10 in saline also showed nanoassemblies by DLS, but with a smaller mean diameter compared to SbL8 (36.3 nm vs. 93.9 nm) (Appendix A).

Knowledge of the pH dependence of stability of these nanoassemblies is critical, as the nanoparticles will go through biological compartments of different pHs to reach their ultimate target. Among those are the acidic gastric fluid (pH 1–2) when nanoassemblies are administered by oral route, the serum with neutral pH and the acidic parasitophorous vacuole (pH 4.5–5.5) where the parasites dwell. The effect of pH on the particle structure and stability was investigated by SAXS and by probing the hydrophobic microenvironment using the lipophilic DPH fluorescent probe (Figure 2).

As shown in Figure 2a, the profile of the SAXS curve in PBS 7.2 differed markedly from that obtained in water (pH 5.5), supporting a conformational change and higher order of aggregation. On the other hand, the SAXS curve in HCl 0.05 M showed a much less structured profile and lower intensity, suggesting disappearance of core–shell nanoassemblies. As illustrated in Figure 2b, the curve of DPH fluorescence as a function of L8 concentration in SbL8 suspensions is notably dependent on pH and ionic strength, demonstrating a strong influence on the critical micellar concentration (CMC). The CMC of SbL8 increased in acidic and low-ionic force media. A lower value of CMC, close to 1 mM, was observed in PBS at neutral pH. Thus, a right shift of the curve demonstrates a lower thermodynamic stability of the nanoassembly, while a left shift means improved stability.

Another important property of the nanoassemblies investigated in this study is their ability to resist dilution and act as drug delivery systems in different pH conditions resembling biological fluids. After incorporation of the DPH lipophilic probe in concentrated solutions of SbL8 and SbL10 in water, the suspensions were diluted below the CMC in media of different pHs (PBS at pHs 4.5, 5.8 or 7.2 and HCl 0.05 M) and the kinetics of fluorescence decay registered (Figure 2c,d).

There was no change in DPH fluorescence intensity as a function of time in PBS at neutral pH, supporting the kinetic stability of the nanoassemblies. On the other hand, the rapid release of DPH was observed at pH below 5.8, with half-time of DPH release lower than 1 h at pH 4.5. Interestingly, DPH was released much more rapidly from SbL8 than SbL10, being the release rate of SbL8 10-fold greater with respect to that of SbL10 at pH 4.5.

### 3.2. Intracellular and Hepatic Accumulation of Sb

A severe pharmacological hurdle for conventional pentavalent antimonials is their low membrane permeability and low intracellular accumulation in the host cell [20]. The amphiphilic character of the new complexes and their ability to self-assemble into nanoparticles is surmised to achieve a higher drug accumulation into the host cell, through increased drug permeation and uptake of the nanoassemblies by phagocytosis. To check for this hypothesis, we compared here the amphiphilic antimony complexes and Glucantime^®^ regarding the intracellular accumulation of Sb in THP-1 macrophage cells. Figure 3a shows the levels of Sb accumulation in these cells after exposure to the amphiphilic complexes and Glucantime^®^ at equal Sb concentrations for 4 h. No detectable Sb accumulated in THP-1 cells from Glucantime^®^ exposition, in contrast to amphiphilic antimony complexes that showed intracellular levels of Sb 20 times higher than Glucantime^®^. Drug accumulation among the different amphiphilic complexes did not show statistically significant difference.

Drug targeting into the liver is critical for the cure of VL, as this organ is a major location for infection. In this context, the influence of the route of administration (oral versus intraperitoneal) and the possible difference between the amphiphilic complexes have a major relevance. Figure 3b shows the hepatic levels of Sb in BALB/c mice 24 h after oral (200 mg Sb/kg) or i.p. (20 mg Sb/kg) administrations of SbL8 or SbL10, in comparison to i.p. Glucantime^®^ at 20 mg Sb/kg. Lower Sb levels in liver were found when amphiphilic complexes were given by oral route, with respect to i.p. route, even with 10-fold higher dose. Strikingly, amphiphilic complexes given intraperitoneally promoted 10-fold higher Sb accumulation levels when compared to Glucantime^®^. Thus, considering the higher accumulation of Sb from amphiphilic antimony complexes, these data demonstrate an effective liver targeting.

### 3.3. In Vitro Antileishmanial Activity and Cytotoxicity

The amphiphilic antimony complexes were further evaluated for their in vitro cytotoxicity and their antileishmanial activities against axenic and intramacrophage *L. donovani* amastigotes, and compared with conventional trivalent and pentavalent antimonial drugs, Milt and AmB (Table 1, Appendix A). Milt and AmB showed in vitro activities with no difference between axenic amastigotes and intramacrophage amastigotes. In contrast Sb(V) derivatives differed widely. This difference is not surprising as it is generally assumed that pentavalent antimonials have to be reduced into trivalent species to exert their activity and that this activation occurs in the host cell. However, it is noteworthy that new pentavalent antimonials (SbL8 and SbL10) exhibited similar in vitro activities as the antimony(III) tartrate on the axenic amastigotes, as evinced in Table 1.

Another striking observation is the much higher activity (at least 40-fold) of SbL10 against the intramacrophage amastigotes, when compared to SbL8 and Glucantime^®^. Moreover, SbL10 is 7 times less cytotoxic than SbL8 against the macrophage cell line used, resulting in a selectivity index as high as 800. This important difference in antileishmanial activity between SbL10 and SbL8 may be related to the difference in the stability of their nanoassemblies and their response to pH changes. Considering that SbL8 and SbL10 promoted similar levels of intracellular Sb accumulation, it is possible that the higher stability of SbL10 nanoassemblies in the parasitophorous vacuole may allow their endocytosis by the parasites and the more effective delivery of the metal to its intracellular target.

### 3.4. In Vitro Study of the Pharmacological Interaction between SbL10 and Milt

Evidence has been obtained previously for the spontaneous incorporation of miltefosine into SbL8 micelles, through increase in the fluorescence intensity of MT-11-BDP miltefosine analog in SbL8 aqueous suspension [5]. A similar behavior was observed with SbL10 suspension (Appendix A), supporting the partitioning of Milt into the hydrophobic core of SbL10 nanoassemblies. The pharmacological interaction between SbL10 and Milt was evaluated in the axenic amastigote and intramacrophage amastigote models, after incorporation of Milt into SbL10 suspension at different ratios. Figure 4 displays the IC_50_ of Milt at increasing proportion of SbL10. The data indicates an increase of the antileishmanial activity with increasing SbL10 proportions to a much higher extent in the intracellular amastigote model than in the axenic model. The conventional chequerboard assay was adapted to calculate the fractional inhibitory concentration of each drug with the modified fixed-ratio method [16,17] (Appendix A). Interpretations of pharmacological effects of drugs in paired combinations were based on FIC values (Appendix A). Results showed that SbL10 and Milt initially combined at 1:4 and 2:3 molar ratios showed no interaction or even an antagonist trend. On the other hand, the 3:2 ratio showed additive effect and 4:1 ratio indicated a synergistic effect, suggesting that incorporation of Milt into SbL10 nanoassemblies at these ratios may favor its delivery to the intracellular environment.

### 3.5. Antileishmanial Efficacy in a Murine VL Model

The antileishmanial efficacies of SbL8 and SbL10 were tested on BALB/c mice infected with *L. donovani*, after treatment either by the oral (at 200 mg Sb/kg/day) or by i.p. (at 20 mg Sb/kg/day) routes. Comparison was made with Glucantime^®^ (at 200 mg/kg/day, i.p. route). Figure 5a,b show, respectively, the parasite loads in the liver and spleen of infected mice after 30 days of treatment.

SbL8 and SbL10 given by i.p. route were equally effective in reducing parasite loads in both liver and spleen, with respect to non-treated control. It is noteworthy that the amphiphilic complexes were as effective as the standard drug Glucantime^®^ given at 10-fold higher dose (200 mg Sb/kg/day). The comparable antileishmanial activities of SbL8 and SbL10 in the liver are in good agreement with the similar levels of hepatic drug accumulation. On the hand, the higher antileishmanial activity of SbL10 in the intramacrophage amastigote model, when compared to SbL8, did not translate into a significant difference in in vivo activity. It is also interesting that the amphiphilic complexes at 20 mg Sb/kg/day by i.p. route showed greater antileishmanial activity than the same complexes given orally at 200 Sb/kg/day.

### 3.6. Antileishmanial Efficacy of Topical Treatment in a Murine CL Model

The antileishmanial activities of SbL8 and SbL10 were further investigated in BALB/c mice infected with *L. amazonensis*, as a model of CL, after topical treatment with a hydrogel containing the amphiphilic complexes at 12% Sb (*w*/*v*). Animals were evaluated according to the lesion growth during treatment and parasite load of the lesion at the end of the treatment. Their effectiveness was compared with respect to non-treated group, as well as to additional groups treated with Glucantime^®^ either topically (hydrogel at 12% Sb (*w*/*v*) Sb) or intraperitoneally (200 mg Sb/kg/day). As shown in Figure 6, topical treatment with SbL10 led to a significant retention of lesion growth, in comparison to the non-treated control, at a level close to that achieved after parenteral treatment with Glucantime^®^. All the antimonial treatments promoted reduction in the parasite load when compared to untreated control, but the difference was only significant for SbL8. The lack of activity of the hydrogel vehicle was confirmed in a separate experiment (Appendix A).

## 4. Discussion

The present study provides new insights into the physicochemical properties and the therapeutic potential of SbL8 and SbL10 nanoassemblies for VL and CL. Previous studies addressed the potential of amphiphilic antimony complexes essentially for oral delivery of Sb [4,5,6]. Here, we moved a step forward by investigating the potential of these nanosystems to carry and deliver Sb(V) and other associated lipophilic substance for parenteral and topical applications.

Fluorescence probing of hydrophobic environment and SAXS measurements demonstrated for the first time the stability of nanoassemblies at neutral pH in aqueous solution, but also the conformational change and release of incorporated lipophilic marker upon medium acidification at pH values close to that of gastric fluid and parasitophorous vacuole. The conformational change as a function of pH is supported by modifications in the SAXS curve profile and DPH partitioning. However, the precise morphological changes in the nanoparticle require further characterization, as we were unable to detect significant difference by TEM between PBS (pH 7.2) (Appendix A) and water [5]. The pH dependence of the stability of the nanoassemblies may be attributed to the acid–base properties of Sb(V). Acidification of the medium is expected to favor protonation of the hydroxyl group of Sb atom, resulting in conformational change of the complexes and their nanoassemblies. Such a protonation is supported by the reduction of the nanoparticle zeta-potential after acidification of the medium (Appendix A). These data are also consistent with previous reports that 1:2 Sb(V)-GMP complex is stable at neutral pH but rapidly dissociates at pH 5 [21]. Interestingly, SbL10 nanoassemblies were found to be more stable than SbL8 ones. This may be attributed to the longer acyl chain of L10 with an enhancement of hydrophobic interactions.

The pH dependence of the stability of these nanoassemblies has important implications for therapeutics. First, it strongly suggests that SbL8 and SbL10 nanosystems release incorporated lipophilic substance in the stomach and are therefore unable to act as a drug carrier by the oral route. On the other hand, these nanoassemblies may be able to carry a lipophilic drug in the bloodstream after parenteral administration, which further release Sb(V) and the co-incorporated drug into the acidic parasitophorous vacuole following phagocytosis. The smaller size of SbL10 nanosystems and their greater stability may afford a better accumulation into the parasite through endocytosis via flagellar pocket. The higher activity of SbL10 against the intramacrophage amastigotes in comparison to axenic amastigotes and the increased activity of Milt after incorporation in SbL10 nanoassemblies (at 2:3 ratio and below) support the specific drug release in the parasitophorous vacuole, rather than nanoparticle endocytosis by the parasite.

The improved targeting of Sb to the host cell (in vitro) and the liver (in vivo) from SbL8 and SbL10 are major findings of the present study. The high intracellular accumulation of Sb can be explained by the amphiphilic character of the complexes (Figure 1) and their self-assembling, in contrast to the marked hydrophilicity of meglumine antimoniate, which may facilitate binding of the nanoassemblies onto the cell surface and their subsequent internalization via phagocytosis. The hydrophobic interaction of the acyl tails of amphiphilic complexes with the lipid membrane may contribute to cell surface binding. Alternatively, the negatively charged polar head of the complexes may interact with receptors on the macrophage surface leading to phagocytosis of the nanosystem. Scavenger receptors are candidates for interaction with amphiphilic Sb complexes as they are known to bind to a range of polyanionic molecules such as phosphatidylserine [22]. Our results are also consistent with previous reports that incorporation of meglumine antimoniate into chitosan-based nanoparticles or liposomes resulted in significantly higher Sb uptake by macrophages and antileishmanial activity, in comparison to the free drug [23,24]. Opsonization is another well-known process that contributes in vivo to rapid clearance of nanoparticulate carriers by macrophages of the mononuclear phagocytic system (MPS) [25]. It consists of the adsorption onto the surface of nanoparticles of opsonins, such as immunoglobulins and complement proteins such as C3, C4, and C5 [26]. Such phenomenon is also expected to contribute to the ability of SbL8 and SbL10 nanosystems to target the liver. Thus, our data supports a scenario in which these Sb nanoassemblies passively target macrophages and organs of the MPS, resulting in higher antileishmanial activity achieved at lower Sb dose in the VL model.

The current work uncovers for the first time the therapeutic potential of SbL8/SbL10 nanoassemblies administered by parenteral route leading to the enhancement of their leishmanicidal effectiveness in comparison to the oral route. Interestingly, it is also the first report on efficacy of amphiphilic Sb(V) complexes on the murine VL model caused by *L. donovani*. The lower cytotoxicity and higher selectivity index of SbL10, regarding the SbL8, suggest a greater potential of SbL10 for treatment of VL. Therapeutic effectiveness was also achieved at lower Sb dose, as compared with the conventional pentavalent antimonials in humans (20 mg Sb/kg/day) equivalent to 240 mg Sb/kg/day in mice. The higher in vitro antileishmanial activity of SbL10 compared to SbL8 in the intramacrophage amastigote model did not translate into significant difference in in vivo activity. As a possible explanation for this apparent discrepancy, the interaction of nanoassemblies with serum components may alter their surface or structural organization and affect their intracellular processing, resulting in either increased delivery of Sb to the parasite from SbL8 or decreased drug delivery from SbL10.

As a main drawback, conventional antimonial therapy is often accompanied by local pain during intramuscular injection and by severe side effects that include cardiotoxicity, pancreatitis, hepatotoxicity and nephrotoxicity [2]. Although toxicity was not addressed in depth in our study, the high drug targeting to the liver and the lower Sb dose required for amphiphilic complexes by i.p. route may result in lower drug accumulation into the heart, pancreas and kidneys and reduced metal-related toxicities in these organs.

Another limitation of conventional antimonial therapy is the high risk of emergence of parasite resistance to antimony [27]. Typical changes observed in Sb-resistant *Leishmania* strains refer to the overexpression of ATP-binding cassette (ABC) transporters responsible for either Sb sequestration inside intracellular vesicles or active extrusion of Sb out of the cells [28]. Down-regulation of the aquaglyceroporin (AQP1) responsible for Sb entry into the parasite has also been reported. As a potential therapeutic benefit of amphiphilic Sb complexes, to be investigated in future studies, the delivery of Sb into the parasite through endocytosis of nanoassemblies may bypass the transporters involved in Sb resistance and keep the cell sensitive to the drug.

Evidence was also obtained for the effectiveness of the topical application of SbL8/SbL10 according to the decrease of the size of the lesion for SbL10 and of the parasite load for SbL8. Nevertheless, the BALB/c mice have an extreme susceptibility to the *L. amazonensis* species used in this study, with the development of increasing, ulcerative, metastatic and difficult-to-treat lesions, which may differ from the course of human CL. In humans, overexuberant cutaneous inflammation is observed in mucosal leishmaniasis which is often refractory to drug treatment [29]. Thus, our experimental CL model may contribute to the moderate activity of the antimonial drugs either applied topically (SbL8, SbL10) or administered parenterally (Glucantime^®^). The activity of the amphiphilic Sb complexes may also be limited by their high affinity for lipid membranes resulting in low skin penetration. Further improvement of the efficacy of topical treatment with these nanoformulations may be achieved by increasing the daily regimen into more usual twice-a-day application and by co-incorporating another active agent in the formulation.

The ability of SbL8 nanoassemblies to incorporate another amphiphilic drug, such as Milt or AmB, and the proposal of combination therapy have been described previously [6]. The present study addresses a gap of knowledge by identifying the most appropriate routes of administration for the use of Sb nanoassemblies as carriers of other lipophilic drugs. Indeed, our work strongly supports the use of parenteral or topical route to guarantee the stability of the nanocarrier and enjoy the full benefit of its targeting properties. Here, we also show that incorporation of Milt in SbL10 suspension at 2:3 ratio (and below) results in highly active formulation in vitro, in the sub-micromolar range. Thus, it would be worth evaluating in future studies the in vivo antileishmanial activity of this specific drug combination by parenteral or topical route. The marked dependence of the pharmacological interaction between SbL10 and Milt upon the drug ratio can be attributed to a drug carrier effect, e.g., the incorporation of Milt in SbL10 micelles, which may predominate at low Milt/SbL10 ratio. Our proposal is that SbL10 nanoassemblies may act as carrier for Milt, therefore enhancing the drug uptake by the host cell or even the parasite. Future studies using the fluorescent Milt analog may help to confirm this model.

Macrophages can be activated to kill intracellular parasites, through increased pro-inflammatory mediator expression (IL-12, IFN-γ and TNF-α) and down-regulation of disease-promoting cytokines (IL-10 and TGF-β). Thus, the progression of the disease, its physiopathology and the effectiveness of drugs including antimonials strongly depend on the host immunologic status [29]. In this context, immunomodulators have been investigated for their efficacy in the treatment of leishmaniasis, with the idea that they may also be used in combination with leishmanicidal drugs [29,30]. Previous studies highlighted therapeutic benefits in CL from the combination of pentavalent antimonials with CpG ODN D35, imiquimod or GM-CSF [31,32,33]. Therefore, one can expect a more effective treatment of CL from the incorporation an immunomodulator in the amphiphilic Sb topical formulation.

## 5. Conclusions

The main conclusions drawn from this study are the stability of SbL8 and SbL10 nanoassemblies at neutral pH in aqueous solution, but the conformational change and release of incorporated lipophilic substance upon acidification. Moreover, SbL8 and SbL10 nanoparticles accumulated at high concentration in macrophages and passively targeted the liver of mice following parenteral administration, resulting in an enhanced antileishmanial activity against VL. We show that incorporation of Milt in SbL10 suspension at 2:3 and 1:4 ratios resulted in a highly active nanoformulation at the sub-micromolar range, in the intramacrophage amastigote model. The high in vitro activities of SbL8 and SbL10 partially translated in vivo in the *L. amazonensis* murine model of CL, either by reducing the growth of lesion size (SbL10) or the parasite load in the lesion (SbL8). This work supports the ability of amphiphilic Sb nanoassemblies to carry lipophilic drugs such as Milt in biological media at neutral pH and to deliver Sb and the co-incorporated drug to host cell and infected tissues. Our study also paved the way for future research on combination therapy using Sb nanoassemblies.

## 6. Patent

The technology described in this article is protected by the Brazilian patent n°: PI 1106237-1, entitled “Nanocarreadores formados por complexos anfifílicos de antimônio(v), processo de obtenção, composições farmacêuticas e uso” from Frézard, F.J.G.; Demicheli, C.P.; Ferreira, W.A.; Fernandes, F.R.; Kato, K.C.; Melo, M.N.; Ferreira, L.A.M., granted on 10/20/2020.

## Figures and Tables

**Figure 1 pharmaceutics-14-01743-f001:**
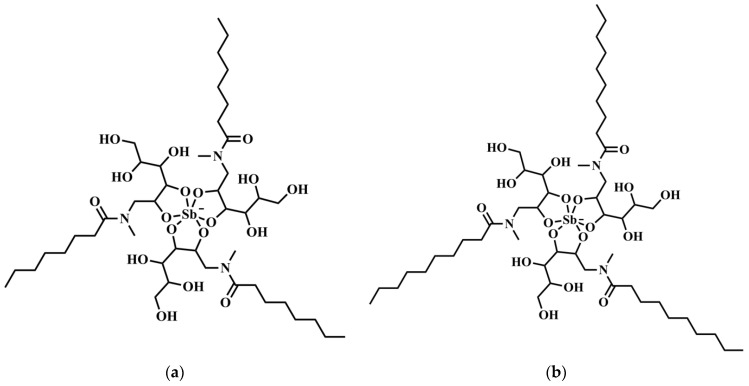
Structural formula of (**a**) SbL8 and (**b**) SbL10 complexes, according to previous chemical characterization [4].

**Figure 2 pharmaceutics-14-01743-f002:**
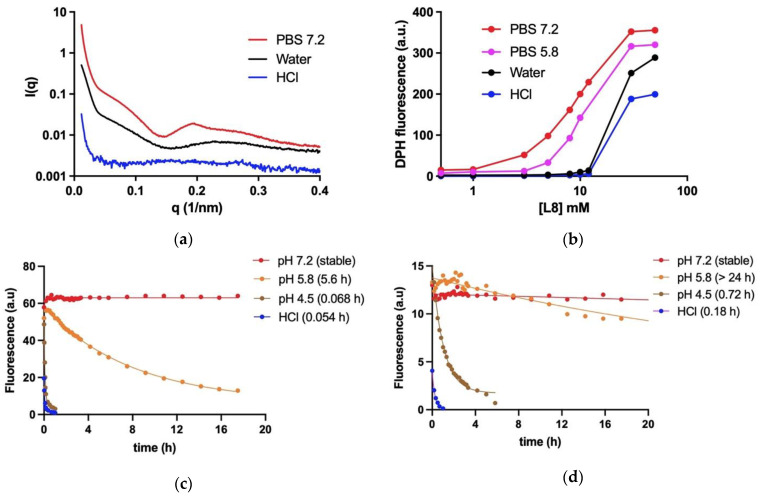
Effect of pH on the thermodynamic and kinetic stabilities of SbL8 and SbL10 nanoassemblies. (**a**) Graph shows SAXS curves at 25 °C of SbL8 at 30 mM L8 in PBS pH 7.2, water (pH 5.5) or HCl 0.05 M. (**b**) DPH fluorescence as a function of L8 concentration in SbL8 suspensions in PBS either at pH 5.8 or 7.2, water (pH 5.5), and HCl 0.05 M. (**c**,**d**) Kinetics of decrease of DPH fluorescence at 37 °C after dilution of DPH-loaded SbL8 and SbL10 suspensions, respectively, in HCl 0.05 M and PBS at pH 4.5, 5.8 and 7.2. Data were fitted according to mono-exponential decay. Half-times of DPH release at different pHs are shown in brackets.

**Figure 3 pharmaceutics-14-01743-f003:**
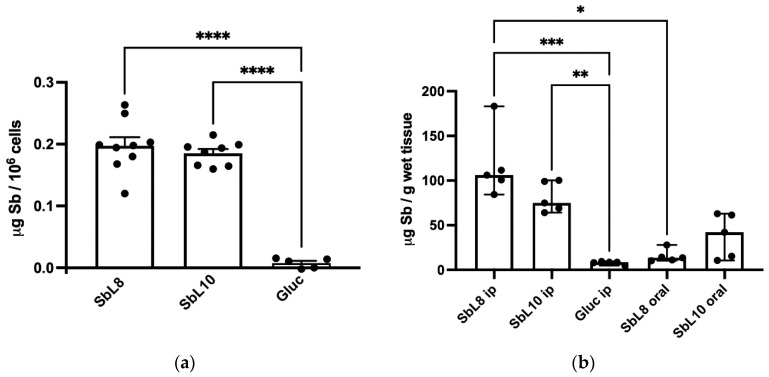
Accumulation of Sb (**a**) in THP-1 cells and (**b**) in the liver of BALB/c mice from SbL8 and SbL10, in comparison to Glucantime^®^ (Gluc). Sb was determined by GFAAS. (**a**) THP-1 cells were incubated for 4 h at 37 °C in the presence of SbL8, SbL10 or Gluc at 50 μg/mL Sb. Data are shown as means of intracellular Sb levels + standard error (*n* = 5–9). **** *p* < 0.0001, one-way ANOVA followed by Dunnett’s multiple comparison post-test. (**b**) BALB/c mice received SbL8 and SbL10 (20 mg Sb/Kg by i.p. route or 200 mg Sb/Kg by oral route) or Gluc (20 mg Sb/Kg, i.p.). The liver was collected after 24 h, homogenized and digested with nitric acid. Data are shown as means ± 95% confidence intervals (*n* = 5). * *p* < 0.05, ** *p* < 0.01; *** *p* < 0.001, Kruskal–Wallis followed by Dunn’s multiple comparison test.

**Figure 4 pharmaceutics-14-01743-f004:**
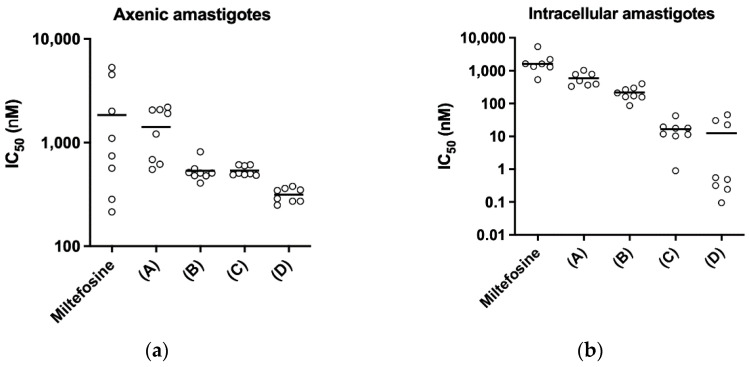
Pharmacological interaction between SbL10 and Milt for *L. donovani*-infected RAW 264.7 cells. Graph displays the change in IC_50_ of Milt when combined with SbL10 at increasing concentrations: (A) = 0.8[Milt] + 0.2[SbL10]; (B) = 0.6[Milt] + 0.4[SbL10]; (C) = 0.4[Milt] + 0.6[SbL10]; (D) = 0.2[Milt] + 0.8[SbL10]). In vitro efficacy assays performed in (**a**) axenic amastigote and (**b**) intramacrophage (RAW 264.7) amastigotes of *L. donovani* (LV9 strain).

**Figure 5 pharmaceutics-14-01743-f005:**
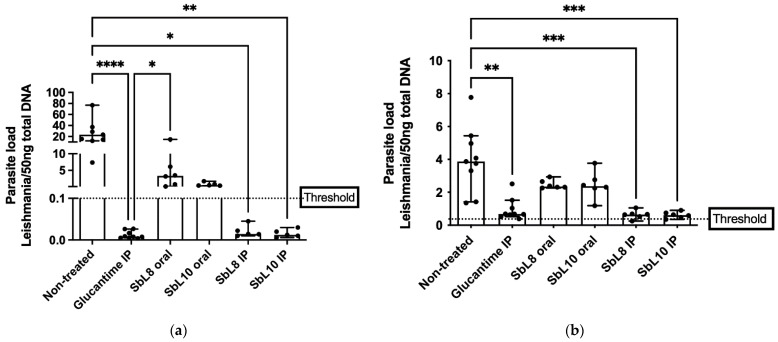
Effectiveness of amphiphilic antimony complexes in *L. donovani*-infected BALB/c mice by oral or i.p. route. A total of 7 days after infection with *L donovani*, mice were treated daily for 30 days with SbL8 or SbL10 by oral (200 mg/kg) or i.p. (20 mg/kg) or with Glucantime^®^ i.p. (200 mg/kg). Parasite loads were determined in the liver (**a**) and spleen (**b**) by qPCR. Data are shown as medians ± 95% confidence intervals (*n* = 6–9). * *p* < 0.05, ** *p* < 0.01; *** *p* < 0.001, **** *p* < 0.0001, Kruskal–Wallis followed by Dunn’s multiple comparison test.

**Figure 6 pharmaceutics-14-01743-f006:**
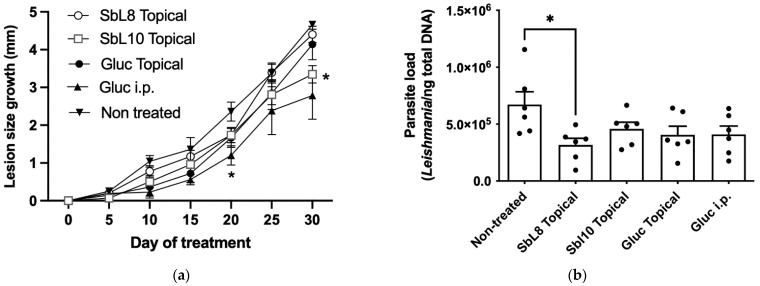
Antileishmanial efficacy of topical treatment with SbL8 and SbL10 nanoformulations in murine CL model. BALB/c mice were infected intradermally with *L. amazonensis* at the tail base. After 35 days of infection, animals (*n* = 6 mice per group) were treated daily for 30 days with either SbL8 hydrogel (12% Sb (*w*/*v*)), SbL10 hydrogel (12% Sb (*w*/*v*), Glucantime^®^ hydrogel (12% Sb (*w*/*v*)) or Glucantime^®^ by i.p. route at 200 mg/kg/day and compared to non-treated control group. (**a**) Variation of the lesion size with time shown as mean ± SEM. * *p* < 0.05 with respect to control untreated group (Two-way repeated measures ANOVA, followed by Tukey’s multiple comparisons test). (**b**) Parasite load as determined by qPCR at the end of treatment. * *p* < 0.05 for comparison of each treated group with the control group, one-way ANOVA test, followed by Dunnett’s multiple comparison post-test. Data are shown as mean + SEM.

**Table 1 pharmaceutics-14-01743-t001:** In vitro cytotoxicity and activity against *L. donovani* axenic and intracellular amastigotes of amphiphilic antimony complexes in comparison to reference drugs and non-complexed ligand. Selectivity index is calculated based on IC_50_ in intramacrophage amastigotes.

Compounds	IC_50_ (µM) in LV9 Amastigotes Like	CC_50_ (µM) in RAW264.7	IC_50_ (µM) in LV9 intra-RAW264.7	* Selectivity Index (SI)
SbL8	286 ± 65	440 ± 35	150 ± 63	2.9
L8	>10,000	~10,000	1750 ± 86	~5.7
SbL10	332 ± 28	2830 ± 100	3.5 ± 0.5	~800
L10	>10,000	>10,000	>10,000	-
Sb(III) tartrate	271 ± 56	N/A	N/A	-
Glucantime^®^	>1200	1050 ± 93	355 ± 11	2.9
Miltefosine	2.2 ± 0.7	>5.0	2.0 ± 0.1	>2.5
Amphotericin B	0.5 ± 0.4	>2.0	0.5 ± 0.1	>4

* Selectivity index is calculated as CC_50_/IC_50_ using IC_50_ in intramacrophage amastigotes.

## Data Availability

Data are contained within the article and Appendix A.

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
