# Peer review of "Nanoassemblies from Amphiphilic Sb Complexes Target Infection Sites in Models of Visceral and Cutaneous Leishmaniases"

_pharmaceutics, 2022, doi:10.3390/pharmaceutics14081743_

Round 1

Reviewer 1 Report

The paper written by the authors describes a technology developed using nano assemblies from amphiphilic Sb complexes. The work is a characterization study of the physical properties of the nanoassmeblies and two routes of administering them for treatment.

The article is well written and the methods are thorough. I have some questions for the authors and things which I think should be clarified in the manuscript.

1. What exactly are the structural differences in the Sb complexes? Is there a image that demonstrates what these particle look like? This would help to understand the graphs better when it is describe that they have conformational changes.

2. Please clarify both what conformational change which is being described and also please describe how to interpret the graphs. This would be especially important for readers who do not use the same characterization systems as described in this manuscript. (i.e. for graphs in figure 1. How do we know that the core shell nano assembly disappeared in HCl solution? --> This could be done by simply saying a right shift demonstrate the degradation of the core nanoassembly while a left shit demonstrates a stabilization of the nanoassembly.

3. Line 437: The Y axis different for these graphs which ch makes it more difficult to compare. Can you elaborate on what is a typical fluorescence score that demonstrates stability of the complexes?

4. Additionally are the lines for figure 2a and 2b in respect to pH at 4.5 and HCl actually different? It seems that since the scale is different on the Y-axis that these lines may actually be the same. Can you clarify the figure?

5. Lines 495-507: A selectivity index of 800 is much higher than all of the values presented. Can you provide a more detailed explanation about how the pH response would explain this result?

Author Response

1. What exactly are the structural differences in the Sb complexes? Is there an image that demonstrates what these particles look like? This would help to understand the graphs better when it is describe that they have conformational changes.

Response: A TEM image of SbL8 particles in PBS is shown in Fig S1a.  We reported previously atomic force microscopy and TEM images of SbL8 particles in water (reference 5). These images together with SAXS data and partitioning of DPH and miltefosine analog in hydrophobic environments support the formation of micelle-like nanoassemblies with a spherical core–shell, as now clearly stated in the Result section (line 419).

2. Please clarify both what conformational change which is being described and also please describe how to interpret the graphs. This would be especially important for readers who do not use the same characterization systems as described in this manuscript. (i.e. for graphs in figure 1. How do we know that the core shell nano assembly disappeared in HCl solution? --> This could be done by simply saying a right shift demonstrate the degradation of the core nanoassembly while a left shift demonstrates a stabilization of the nanoassembly.

Response: “The conformational change as a function of pH is supported by the modifications in SAXS curve profile and DPH partitioning. But the precise morphological changes in the nanoparticle require further characterization, as we were unable to detect significant difference by TEM between PBS (pH 7.2) and water.” This comment was added to the Discussion section (lines 612-615).
Furthermore, the description of the results was improved (lines 428-430), as recommended by the Reviewer.

3. Line 437: The Y axis different for these graphs which ch makes it more difficult to compare. Can you elaborate on what is a typical fluorescence score that demonstrates stability of the complexes?

Response: The difference in fluorescence scale of graphs a and b (Fig 2) can be attributed to a difference in DPH quantum yield between SbL8 and SbL10 nanoassemblies, presumably related to structural differences. The stability of the complex is related to the fluorescence kinetic profile (rather than the absolute fluorescence intensity). Thus, the lack of fluorescence change as a function of time means stability of the nanoassemblies. This point was clarified in the Result section (lines 449-451).

4. Additionally are the lines for figure 2a and 2b in respect to pH at 4.5 and HCl actually different? It seems that since the scale is different on the Y-axis that these lines may actually be the same. Can you clarify the figure?

Response: Please, see our response to point 3 and consider that the dissociation half-times shown in Table 2 (as determined from the curves) inform about the stability of DPH incorporation in nanoassemblies. From Table 2 data, we observe that SbL8 at pH 4.5 and SbL10 in HCl showed significantly different half-times (0.068 h vs 0.18 h, respectively).

5. Lines 495-507: A selectivity index of 800 is much higher than all of the values presented. Can you provide a more detailed explanation about how the pH response would explain this result?

Response: As a possible explanation, the higher stability of SbL10 nanoassemblies in the parasitophorous vacuole may allow their endocytosis by the intracellular parasite and the more effective delivery of the metal to its intracellular target. This explanation was added to the Result section (lines 513-515). 

The authors thank the Reviewer for his valuable recommendations that help to improve our manuscript.

Reviewer 2 Report

This work is a continuation of previous works where the authors described the dispersion of Sb in a surfactant. This work intends to focus on the biological utility of these dispersions under in vitro and in vivo settings. The authors claim that the formulations tested do penetrate cells more effectively than the common medication of Sb,  Glucantime®

The authors show a significant increase in the penetration of their nanoparticles into cells, however, in the in vivo experiments, the improvement is less significant. 

The article is not focused and contains non-relevant characterization parameters of the formulation. The authors are requested to focus only on the cell accumulation and in vivo data and leave out the in vitro characterization which is less relevant to the focus of the article. 

Author Response

Response:
The authors consider that the physicochemical characterization of the nanoformulations brings important insight into their mode of action and the most promising routes for the administration of the nanoassemblies as carriers of other lipophilic drugs. The report of the pH-sensitivity of the nanoassemblies and the differences between SbL8 and SbL10 are especially interesting and original. Removing this part would leave our work much less relevant and interesting to the readers.
Therefore, the authors prefer to maintain their manuscript in its current format. 

Reviewer 3 Report

Known in the field based on previous literatures:

1.  Leishmaniasis, is caused by protozoan parasites which are transmitted by the bite of infected female phlebotomine sandflies. Basically, parasites existed in two form- amastigote form which is found in the mononuclear phagocytes and circulatory systems of humans while extracellular and motile promastigote form is found in the alimentary tract of sandflies.

2.  The use of nanoparticles and their carriers in drug delivery and treatment of leishmaniasis is reported and used globally.

3.  Combination oral therapy against Leishmania infection in BALB/c mice using nanoassemblies made from amphiphilic antimony complex incorporating other drugs like miltefosine and amphotericin showed significant antileishmanial activity. 

In this manuscript authors performed and reported following findings:

1.  Authors tested the hypothesis whether Sb nanoassemblies favor drug targeting of infection sites after parenteral or topical application. They evaluated the stability of these nanoassemblies at different pH, accumulation of Sb in the host, and their activities in models of visceral and cutaneous leishmaniasis.

2. Authors synthesized amphiphilic antimony complexes and evaluated the incorporation of miltefosine. Further, authors physiochemically characterized the SbL8 and SbL10 and measured the accumulation of intracellular antimony. 

3.     Authors also checked the antileishmanial activity of above drugs and their in vitro cytotoxic effect. 

Minor Concerns:

Although, authors performed and explained nicely about the nanoassemblies and their role in antileishmanial activity still there are many questions. The following minor suggestions if incorporated could help in the better understanding of the significance of the work and implications. Some of the minor concerns are-

1.  Combination oral therapy against leishmania infection in BALB/c mice using nanoassemblies made from amphiphilic antimony complex incorporating drugs like miltefosine, amphotericin B, or other are already existed. What does it add to the subject area compared with published material excluding the stability of SbL8 and SbL10 nanoassemblies at neutral pH and their release? Please explain how does it address a specific gap in the field? 

2.  Explain how your study is different from rest if the drug still shows cytotoxicity? Please discuss bit more about the drawbacks such as toxicities and the emergence of drug resistance and their regulatory proteins.

3.   Discussion could be improved, and authors should discuss bit more about the parasite’s burdens and their effect on liver and spleen in terms of color, shape, and the importance of immunological cells which control the parasites growth in the presence of SbL8 and SbL10 in relation to saline-treated control group.

4.  Have you checked the host immune response generated in mice treated with parasites in presence/ absence of these drugs? Since, immune response regulates both host as well as parasites viability.

5.   Is it physiologically possible to use both SbL8 and SbL10 together to see the additive or synergistic effect of drugs as compared to alone?

6.   What are the reasons of in vitro SbL10 antileishmanial activity variation in the L. donovani murine VL after parenteral administration and moderate activity in the L. amazonensis murine CL after topical treatment?  

Author Response

1.  Combination oral therapy against leishmania infection in BALB/c mice using nanoassemblies made from amphiphilic antimony complex incorporating drugs like miltefosine, amphotericin B, or other are already existed. What does it add to the subject area compared with published material excluding the stability of SbL8 and SbL10 nanoassemblies at neutral pH and their release? Please explain how does it address a specific gap in the field? 

Response: Thank you for this valuable recommendation. To address the Reviewer concern, the following comments were added to the Discussion of the revised manuscript (lines 687-689):
The present study addresses a gap of knowledge by identifying the most appropriate routes of administration for the use of Sb nanoassemblies as carriers of other lipophilic drugs. Indeed, our work strongly supports the use of parenteral or topical route to guarantee the stability of the nanocarrier and get full benefit of its targeting properties.

2.  Explain how your study is different from rest if the drug still shows cytotoxicity? Please discuss bit more about the drawbacks such as toxicities and the emergence of drug resistance and their regulatory proteins.

Response: Thank you for this valuable recommendation. To address the Reviewer concerns, the two following paragraphs regarding toxicities and drug resistance were added to Discussion (lines 658-660 and lines 664-669):

“As a main drawback, conventional antimonial therapy is often accompanied by local pain during intramuscular injection and by severe side effects that include cardiotoxicity, pancreatitis, hepatotoxicity and nephrotoxicity [2].”

“Another drawback of conventional antimonial therapy is the high risk of emergence of parasite resistance to antimony [22]. Typical changes observed in Sb-resistant Leishmania strains refer to the overexpression of ATP-binding cassette (ABC) transporters responsible either for Sb sequestration inside intracellular vesicles or for active extrusion of Sb out of the cells [23]. Downregulation of the aquaglyceroporin (AQP1) responsible for Sb entry into the parasite has also been reported.”

[22] Ponte-Sucre, A.; Gamarro, F.; Dujardin, J.C.; Barrett, M.P.; Lopez-Velez, R.; Garcia-Hernandez, R.; Pountain, A.W.; Mwenechanya, R.; Papadopoulou, B. Drug resistance and treatment failure in leishmaniasis: a 21st century challenge. PLoS Negl Trop Dis 2017, 11, e0006052, doi:10.1371/journal.pntd.0006052.
[23] Frézard, F.; Monte-Neto, R.L.; Reis, P.G. Antimony transport mechanisms in resistant leishmania parasites. Biophys. Rev. 2014, 6, 119–132, doi:10.1007/s12551-013-0134-y.

We also included additional comments (lines 660-663 and lines 669-672) on the potential therapeutic benefits of using amphiphilic Sb complexes, regarding both aspects.

3.   Discussion could be improved, and authors should discuss bit more about the parasite’s burdens and their effect on liver and spleen in terms of color, shape, and the importance of immunological cells which control the parasites growth in the presence of SbL8 and SbL10 in relation to saline-treated control group.

Responses: The importance of immunological cells in the control of parasite growth and how these cells can be explored to improve the treatment of leishmaniasis is now commented in the Discussion of the revised manuscript as follows (lines 700-710):
“Macrophages can be activated to kill intracellular parasites, through increased pro-inflammatory mediators expression (IL-12, IFN-γ and TNF) and down-regulation of disease-promoting cytokines (IL-10 and TGF-β). Thus, the progression of the disease, its physiopathology and the effectiveness of drugs including antimonials strongly depend on the host immunologic status [24]. In this context, immunomodulators have been investigated for their efficacy in the treatment of leishmaniasis, with the idea that they may also be used in combination with leishmanicidal drugs [24,25]. Previous studies highlighted therapeutic benefits from the combination of pentavalent antimonials with CpG ODN D35, imiquimod or GM-CSF [26-28]. Therefore, one can expect a more effective treatment of CL from the incorporation an immunomodulator in the amphiphilic Sb topical formulation.”

24.    Novais, F.O.; Amorim, C.F.; Scott, P. Host-directed therapies for cutaneous leishmaniasis. Front. Immunol. 2021, 12, 660183, doi:10.3389/fimmu.2021.660183.
25.    Roatt, B.M.; Aguiar-Soares, R.D.O.; Coura-Vital, W.; Ker, H.G.; Moreira, N.D.; Vitoriano-Souza, J.; Giuchetti, R.C.; Carneiro, C.M.; Reis, A.B. Immunotherapy and immunochemotherapy in visceral leishmaniasis: Promising treatments for this neglected disease. Front. Immunol. 2014, 5, 272, doi:10.3389/fimmu.2014.00272.
26.    Thacker S.G.; McWillams, I.L.; Bonnet, B.; Halie, L.; Beaucage, S.; Rachuri, S.; Dey, R.; Duncan, R.; Modabber, F.; Robinson, S.; et al. CpG ODN D35 improves the response to abbreviated low-dose pentavalent antimonial treatment in nonhuman primate model of cutaneous leishmaniasis. PLoS Negl. Trop. Dis. 2020, 14, e0008050, doi:10.1371/journal.pntd.0008050.
27.    Arevalo, I.; Tulliano, G.; Quispe, A.; Spaeth, G.; Matlashewski, G.; Llanos-Cuentas, A.; Pollack, H. Role of imiquimod and parenteral meglumine antimoniate in the initial treatment of cutaneous leishmaniasis. Clin. Infect. Dis. 2007, 44, 1549-1554, doi:10.1086/518172.
28.    Almeida, R.P.; Brito, J.; Machado, P.L.; De Jesus, A.R.; Schriefer, A.; Guimarães, L.H.; Carvalho, E.M. Successful treatment of refractory cutaneous leishmaniasis with GM-CSF and antimonials. Am. J. Trop. Med. Hyg. 2005, 73, 79–81, PMID:16014838.

4.  Have you checked the host immune response generated in mice treated with parasites in presence/ absence of these drugs? Since, immune response regulates both host as well as parasites viability.

Response: Since antimonial drugs are not expected to modulate the host immune response, we did not investigate this specific aspect. But, as we are dealing with new ligands, a distinct pharmacological profile may emerge, therefore we agree with the Reviewer that this would be interesting.
In fact, as described in our response to point 3 (and commented in the Discussion), we expect benefits from the combination of the new amphiphilic antimonials with immunomodulators.

5.   Is it physiologically possible to use both SbL8 and SbL10 together to see the additive or synergistic effect of drugs as compared to alone?

Response: Drugs with distinct mechanisms of action are usually preferred for investigation of synergistic effect. SbL8 and SbL10 have the same active agent (the metal) and, most probably, present the same mechanism of action. 

6.   What are the reasons of in vitro SbL10 antileishmanial activity variation in the L. donovani murine VL after parenteral administration and moderate activity in the L. amazonensis murine CL after topical treatment?  

Responses: Our explanation for the moderate activity in the L. amazonensis murine CL after topical treatment has been complemented in the Discussion as follows (lines 677-681):  “Nevertheless, the BALB/c mice have an extreme susceptibility to the L. amazonensis species used in this study, with the development of increasing, ulcerative, metastatic and difficult-to-treat lesions, that may differ from the course of human CL. In humans, overexuberant cutaneous inflammation is observed in mucosal leishmaniasis which is often refractory to drug treatment [23]. Thus, our experimental CL model may contribute to the moderate activity of the antimonial drugs either applied topically (SbL8, SbL10) or administered parenterally (Glucantime®).”

A possible explanation for the in vitro SbL10 antileishmanial activity variation in the L. donovani murine VL after parenteral administration and moderate activity was added to the Discussion as follows (lines 647-653):
“The higher in vitro antileishmanial activity of SbL10 compared to SbL8 in the intramacrophage amastigote model did not translate into significant difference in in vivo activity. As a possible explanation for this apparent discrepancy, the interaction of nanoassemblies with serum components may alter their surface or structural organization and affect their intracellular processing, resulting in either increased delivery of Sb to the parasite from SbL8 or decreased drug delivery from SbL10.”

The authors thank the Reviewer for his valuable comments that help to improve greatly our manuscript.

Reviewer 4 Report

The manuscript “Nanoassemblies from amphiphilic Sb complexes target infection sites in models of visceral and cutaneous leishmaniases” explores the potential of amphiphilic prodrugs SbL8 and SbL10 for the treatment of Leishmania infections through both topical and parenteral routes. The authors present a thorough in vitro study as well as in vivo proof-of-concept studies in relevant mice models. The study provides insight on the most efficient administration route, for example discarding the oral route due to poor stability of the nanoassemblies at low pH. Additionally, the results show significant differences between both candidates in the treatment of VL or SL, probably explained by structural parameters of each nanoassembly. Considering the relevance of Leishmania infections and the thorough study herein presented, I suggest publishing this manuscript as Regular Article in Pharmaceutics in the present form. This manuscript will be of interest for Pharmaceutics readers.

Author Response

The authors thank the Reviewer for his kind comments.

Round 2

Reviewer 2 Report

This article is a continuation of previous works that deal with the same compounds. The data presented has some addition to what already was published, although the activity of the compounds remain marginal. This article can be reduced in size and focus on the new findings, the targeting aspect. The authors should add the structures of the compounds and explain how and why these structures provide a targeting effect. Also, the authors should explain the activity data which does not present an improvement in activity. 

Author Response

The authors would like first to apologize for their incomplete response to the Reviewer in the first round of the review process.

  1. The data presented has some addition to what already was published, although the activity of the compounds remain marginal.

Response: we agree that the activity is marginal for cutaneous leishmaniasis. However, the activity of amphiphilic Sb complexes is significant by parenteral route in visceral leishmaniasis, considering the targeting effect and the high antileishmanial activity achieved at much lower dose of Sb, when compared to Glucantime (20 mg Sb/kg/day vs. 200 mg Sb/kg/day).

  1. This article can be reduced in size and focus on the new findings, the targeting aspect.

Response: We agree with the Reviewer that the targeting aspect represents new and important findings. To address the Reviewer concern, the initial part related to the characterization of particle size and zeta potential was removed and the text related to the influence of pH on the stability of the nanoassemblies was condensed and a single figure was maintained (instead of two figures and one Table).

  1. The authors should add the structures of the compounds and explain how and why these structures provide a targeting effect.

Response: The authors thank the Reviewer for the valuable recommendation.

The structures of SbL8 and SbL10 complexes are now presented in Figure 1.

The Discussion was improved regarding the possible mechanisms involved in the targeting effect. Please, see the new paragraph from line 688 to line 709 (marked version).

The following references were also added to support our interpretation of the results.

  1. Tempone, A.G.; Perez, D.; Rath, S.; Vilarinho, A.L.; Mortara, R.A.; de Andrade, H.F. Jr. Targeting Leishmania (L.) chagasiamastigotes through macrophage scavenger receptors: the use of drugs entrapped in liposomes containing phosphatidylserine. Antimicrob. Chemother. 2004, 54, 60-68, doi:10.1093/jac/dkh281.
  2. Sarwar, H.S.; Ashraf, S.; Akhtar, S.; Sohail, M.F.; Hussain, S,Z.; Rafay, M.; Yasinzai, M.; Hussain, I.; Shahnaz, G. Mannosylated thiolated polyethylenimine nanoparticles for the enhanced efficacy of antimonial drug against leishmaniasis. Nanomedicine (Lond) 2018, 13, 25-41, doi:10.2217/nnm-2017-0255.
  3. Borborema, S.E.; Schwendener, R.A.; Osso, J.A. Jr.; de Andrade, H.F. Jr.; do Nascimento, N. Uptake and antileishmanial activity of meglumine antimoniate-containing liposomes in Leishmania (Leishmania) major-infected macrophages. J. Antimicrob. Agents 2011, 38, 341-347, doi:10.1016/j.ijantimicag.2011.05.012.
  4. Fam, S.Y.; Chee, C.F.; Yong, C.Y.; Ho, K.L.; Mariatulqabtiah, A.R.; Tan, W.S. Stealth coating of nanoparticles in drug-delivery systems. Nanomaterials (Basel) 2020, 10, 787, doi: 10.3390/nano10040787.
  5. Moghimi, S.M.; Patel, H.M. Serum-mediated recognition of liposomes by phagocytic cells of the reticuloendothelial system----the concept of tissue specificity. Drug Deliv. Rev. 1998, 32, 45-60, doi:10.1016/s0169-409x(97)00131-2.

  1. Also, the authors should explain the activity data which does not present an improvement in activity. 

The data in the cutaneous leishmaniasis model did not show high improvement in activity for SbL8 and SbL10. It also noteworthy that conventional i.p. treatment with standard drug Glucantimeâ at dose as high as 200 mg Sb/kg/day also showed low efficacy.

As stated in the Discussion, from line 744 to 750 (p. 16 – marked version), one explanation for this result may be the severity of the inflammatory response of BALB/c mice to L. amazonensis, resulting in difficult-to-treat lesions. Another explanation was added to Discussion (line 750-751) as follows: “The activity of the amphiphilic Sb complexes may also be limited by their high affinity for lipid membranes resulting in low skin penetration.”

The authors thank the Reviewer for his valuable comments that help to improve greatly our manuscript.